# Impact of Altered Gut Microbiota and Its Metabolites in Cystic Fibrosis

**DOI:** 10.3390/metabo11020123

**Published:** 2021-02-22

**Authors:** Aravind Thavamani, Iman Salem, Thomas J. Sferra, Senthilkumar Sankararaman

**Affiliations:** 1Department of Pediatrics, Division of Pediatric Gastroenterology, UH Rainbow Babies & Children’s Hospital, Case Western Reserve University School of Medicine, Cleveland, OH 44106, USA; Aravind.Thavamani@UHhospitals.org (A.T.); Thomas.Sferra@UHhospitals.org (T.J.S.); 2Center for Medial Mycology, Case Western Reserve University School of Medicine, UH Cleveland Medical Center, Cleveland, OH 44106, USA; ims31@case.edu

**Keywords:** gut microbiota, microbiome, diversity, cystic fibrosis, dysbiosis, probiotics, metabolites

## Abstract

Cystic fibrosis (CF) is the most common lethal, multisystemic genetic disorder in Caucasians. Mutations in the gene encoding the cystic fibrosis transmembrane regulator (CFTR) protein are responsible for impairment of epithelial anionic transport, leading to impaired fluid regulation and pH imbalance across multiple organs. Gastrointestinal (GI) manifestations in CF may begin in utero and continue throughout the life, resulting in a chronic state of an altered intestinal milieu. Inherent dysfunction of CFTR leads to dysbiosis of the gut. This state of dysbiosis is further perpetuated by acquired factors such as use of antibiotics for recurrent pulmonary exacerbations. Since the gastrointestinal microbiome and their metabolites play a vital role in nutrition, metabolic, inflammatory, and immune functions, the gut dysbiosis will in turn impact various manifestations of CF—both GI and extra-GI. This review focuses on the consequences of gut dysbiosis and its metabolic implications on CF disease and possible ways to restore homeostasis.

## 1. Introduction

Cystic fibrosis (CF) is the most common autosomal recessive disease among Caucasians, affecting more than 30,000 patients in the United States and almost 2.5 times this number worldwide [1]. It is characterized by mutations in the gene coding for the cystic fibrosis transmembrane regulator (CFTR) protein. CFTR dysfunction results in impairment of epithelial ion transport (Cl^−^ and HCO_3_^−^) and impaired mucus hydration [2]. Patients with CF have multisystemic involvement characterized by frequent pulmonary exacerbations and various extrapulmonary manifestations involving the gastrointestinal (GI) tract, pancreas, liver, reproductive organs, and skin [3,4]. Recurrent pulmonary infections are the major cause of morbidity and mortality in CF, and often require frequent hospitalizations for management with prolonged courses of antibiotics. Progressive pulmonary involvement can also result in nutritional deficiency related to a decrease in appetite and increased caloric needs from inflammatory catabolism. GI symptoms of CF are often secondary to mucous inspissation and intestinal dysmotility that may manifest as meconium ileus in the newborn period and present with gastroesophageal reflux disease, small intestinal bacterial overgrowth, distal intestinal obstruction syndrome, and constipation later in life.

Microorganisms reside in all mucosal surfaces of the host. The microbiome refers to the collective genes harbored by these microorganisms. As the gut harbors the highest number and most diverse microbiota, gut dysbiosis and its implications have been increasingly recognized in many conditions [5,6,7,8]. Microbial diversity is strongly correlated with healthy status and diversity takes both richness (number of microbial species identified in an ecosystem) and evenness (relative abundance of each represented species) into consideration [7,8]. Reduced gut microbial diversity is observed in several inflammatory, metabolic, immune-related, and systemic diseases [8,9,10]. Non-homeostatic imbalance in the composition, diversity, or function of these resident microorganisms is termed dysbiosis [7,11,12,13,14,15].

Host–microbe interactions impact various host entities such as immune function, metabolic pathways, growth, and development, yet many of the mechanisms involved in these interactions remain elusive [8,16]. The abnormal GI milieu prevailing in CF is associated with dysbiosis. Despite increasing recent interest in microbial research in CF, the temporal changes and clinical significance of gut dysbiosis and its effects on the disease process remain poorly understood [6]. About 80% of gut microorganisms cannot be cultured by standard culture methods [14,17]. Technologic advancements now permit accurate profiling of an individual’s microbiome composition. The advent of culture-independent molecular methods such as high-throughput next-generation sequencing of the 16 small ribosomal subunit RNA (16S rRNA) deepened our understanding of the gut microbiome, and more sophisticated techniques such as metagenomics, metatranscriptomics, proteomics, and metabolomics, collectively referred to as multiomics, have helped us to understand both taxonomic and functional aspects of dysbiosis in detail [4,5,7,11,14]. The application of metagenomics to the culture methods, referred to as culturomics, has extended our insights in further recognizing the relevant microbiota [11]. This review focuses on the implications of gut dysbiosis in CF (Figure 1) and possible ways to restore homeostasis.

## 2. Normal Gastrointestinal Microbiota

### 2.1. Normal Gut Microbiota Composition

Microorganisms inhabiting the gut include bacteria, archaea, viruses, and fungi [7,8]. Bacteria are the most studied gut microbes, and the term microbiota is often used synonymously to refer to bacteria [5,11]. The microbiome contains approximately five million genes and is two-order times higher than the 23,000 genes in the human genome [8,10]. Normal infants have a unique microbial composition; however, by 2–3 years of life, the microbiota more closely resembles that of the adult gut, remaining relatively stable without further perturbations [7,18]. Among the bacterial phyla, Firmicutes and Bacteroidetes predominate in healthy adults and constitute about 90–95%, followed by Proteobacteria and Actinobacteria [7,8,13,17]. Other less-frequently represented phyla include Fusobacteria, Verrucomicrobia, and Spirochaetes [7,13]. The concentration of commensal bacterial microbiota increases along the GI tract, reaching a maximum of approximately 10^12^ cells/gm in the colon [10,11,13]. In the GI tract, there also exists a lumen-to-surface gradient, with most species populating the lumen, and some (such as *Akkermansia muciniphila* and many species of Proteobacteria) are more specialized to reside in the mucosal intestinal layer [10,11]. Most species prevail in the anaerobic environment of the colon.

### 2.2. Factors Affecting Gut Microbiota

The GI microbial milieu is a result of complex dynamic host–microbial interactions of multiple physicochemical factors, such as nutrient availability, anaerobic environment, antimicrobial secretions such as gastric acid and bile, and normal motility of the GI tract [8]. Additionally, the host-specific factors such as genetic composition, dietary changes, exposure to smoking, and medications such as antibiotics further shape the GI microbiota [7,10,11,14,16,19,20,21]. David et al. reported reproducible substantial and instantaneous alterations in the intestinal microbiome even with short-term diet changes, an observation that was further supported by Carmody et al. in mouse models [22,23]. Elegant work by De Filippo et al. highlighted fecal microbiota in children from Europe and rural Africa as genetically well-adapted, particularly for metabolizing the nutrients in their respective diets [24]. Furthermore, this profound impact of diet can be clearly appreciated when infants are switched from breastfeeding to complementary foods, which helps the maturation of their gut microbiome during the first year of life [8,25]. Similarly, antibiotic treatment has a profound and often long-lasting impact on the GI microbial composition, often resulting in a postantibiotic dysbiotic state [8,25].

### 2.3. Mutualistic Functions of Gut Microbiota

Most microbial residents are mutualistic to the host. Microbiome and host interactions are mediated via multiple ways, including the microbial-associated molecular patterns (popularly referred to as the MAMPS) or via several proteins and metabolites expressed by microbial activity, which influence the host via paracrine and endocrine mechanisms [8,10,14,16]. In healthy individuals, the gut microbiome plays a vital role in host homeostasis, including prevention of pathogenic infections, supply of essential vitamins, xenobiotic metabolism, energy source via production of short-chain fatty acids (SCFAs), biotransformation of conjugated bile acids, regulation of host metabolism, metabolism of xenobiotics, and elicitation of mucosal immune responses (Table 1 and Figure 2) [5,7,8,13,16,26,27]. SCFAs include acetate, butyrate, and propionate, which are produced by anaerobic colonic microbiota as a result of fermentation of indigestible carbohydrates [16,28,29]. The prominent SCFA producers include genera *Eubacterium, Faecalibacterium, Ruminococcus**,* and *Roseburia,* belonging to the order Clostridiales (phylum Firmicutes) and many species of phylum Bacteroidetes [28,29]. SCFAs, importantly butyrate, are crucial for maintaining the intestinal epithelial maturation and homeostasis, colonic sustenance, preservation of epithelial integrity, amelioration of mucosal inflammation, maintenance of colonic pH, and regulation of intestinal motility [2,9,10,27,29,30,31].

## 3. Gut Dysbiosis in Cystic Fibrosis

### 3.1. Mechanisms of Gut Dysbiosis in CF

The intestinal tract involvement in CF begins during fetal life and is noted in most patients [2,32]. Associated common GI symptoms in CF include abdominal pain, bloating, distention, steatorrhea, poor weight gain, and constipation [2,33]. CFTR is expressed in intestinal epithelial cells, and its dysfunction in CF results in defective CFTR protein, which leads to various physiological and biochemical imbalances. These include thick and inspissated mucus due to chloride channel dysfunction, defective bicarbonate secretion altering the intestinal pH milieu, prolonged intestinal transit, pancreatic insufficiency, enhanced intestinal inflammation, and altered immune mechanisms with an impaired intestinal barrier function [9,34,35,36,37,38,39,40,41] (Table 2). Clinically, the GI symptoms in CF are mainly related to obstruction, malabsorption, and inflammation [2,33]. The changes in the GI ecosystem result in an early and chronic state of gut dysbiosis. Although this dysbiosis is associated primarily with CFTR dysfunction, several acquired factors, such as repeated antibiotic exposures, high-calorie diets, and other medications (acid-suppressive medications, opioids, anticholinergic agents, posttransplant immunosuppressive medications, etc.), may further perpetuate the gut dysbiosis in CF [18,30,40,42,43]. (Figure 1)

Meeker et al. demonstrated that CFTR dysfunction actively modulates the gut microbiome composition [33]. Using germ-free CF and germ-free non-CF mouse models, fecal microbiota transplantation (FMT) was carried out with fecal inoculum from select pathogen-free mice [33]. Hierarchical analysis based on the relative abundance of the microbiota revealed a distinct microbial pattern due to CFTR dysfunction [33]. Schippa et al. analyzed the effect of different CFTR gene variants and their association with intestinal dysbiosis [44]. A higher degree of gut dysbiosis was observed among patients with severe phenotypic expression and homozygous delF508 mutations, where the predominant groups are *Escherichia coli* (*E. coli)* and *Eubacterium biforme,* with relative depletion of *Bifidobacterium* and *Faecalibacterium* species [44]. Microbial representation varied based on whether the patient had one or two alleles with delF508 mutations or other CF-related mutations [44]. A negative correlation was also noted between the utilization of antibiotics and intestinal microbial diversity [45]. In a two-year longitudinal follow-up study, Kristensen and colleagues demonstrated that dysbiosis in CF infants is partly due to antibiotics and antibiotic exposure within the first month of life correlated with reduced alpha diversity (within-sample individual diversity, characterized by the richness and/or evenness of species distribution) [15,37].

In approximately 85% of CF patients, exocrine pancreatic insufficiency is present, which significantly impacts the absorption of fat and fat-soluble vitamins, necessitating lifelong pancreatic enzyme replacement therapy [46]. The correlation between pancreatic insufficiency and gut dysbiosis is controversial, and further large-scale studies are needed. For example, Nielsen et al. reported that gut dysbiosis was relatively less pronounced in pancreatic-sufficient CF patients compared to patients with pancreatic insufficiency [18]. However, studies by Burke and colleagues did not reveal significant microbial disparities in diversity or abundance in CF patients based on their pancreatic exocrine function [45]. Along with fat malabsorption, the high-calorie, high-fat diet prescribed for CF patients may also predispose them to gut dysbiosis [30]. Mice fed a high-fat diet exhibited gut dysbiosis [47]. Furthermore, Matamouros et al. showed that *E. coli* fecal isolates from CF patients differed from those without CF, as they showed abundant growth in glycerol medium, which is likely an adaptation due to selective pressure from the increased availability of intestinal fats in CF [48].

### 3.2. Patterns of Gut Dysbiosis in CF

#### 3.2.1. Decrease in Microbial Diversity 

Both pediatric and adult studies demonstrate that the gut microbiota in CF differs taxonomically and functionally from healthy individuals with reduced abundance and species richness, and this distinction begins early in life [18,42,43,49]. Many studies are observational in methodology, involving a small number of CF patients and healthy controls, and of those, only a few are longitudinal (Table 3). Stool samples are utilized in majority of studies, and few evaluated the colonic mucosal microbiome. A consistent finding is decreased microbial diversity in CF patients compared to healthy controls [18,30,39,45,50,51]. However, there is controversy as to whether more deviation in the abundance and diversity of operational taxonomic units (OTUs) is observed with progression of age [11,18,52]. As an example, Nielsen et al. illustrated that OTU abundance was significantly lower in CF patients in contrast to control, and that species abundance among 15-year-old CF patients was lower than in a 1-year-old in the control group [18]. Vernocchi et al. described a distinct enterophenotype characteristic of CF patients that was age-independent, and Manor et al. reported that taxonomic and functional microbial changes in young children with CF decreased with the progression of age [39].

#### 3.2.2. Alteration of Gut Microbial Composition

Along with reduced gut microbial diversity, there is a significant alteration of microbial composition in patients with CF [9,26,30,37,39,40,50,55] (Table 4 and Figure 1), For example, an increase in Firmicutes, a reduction in Bacteroidetes, along with a higher abundance of pro-inflammatory microbiota such as those belonging to Enterobacteriaceae, *Streptococcus,* and *Veillonella* are consistently reported in patients with CF [6,45]. Metagenomic analysis revealed that the hallmark of gut dysbiosis in CF is the significant increase in γ-Proteobacteria concentration with the predominant order Enterobacterales. Enterobacteriaceae-dominant dysbiosis driven by a higher relative abundance of *E. coli* species has been frequently observed in CF and often noted early in life [40,56]. Manor et al. also reported an increase in the species *Enterococcus faecalis and Enterococcus faecium*, known to frequently exhibit antibiotic resistance [30]. In addition, several studies note that CF patients have relative depletion of genus *Bifidobacterium* and *Clostridium* compared to their healthy siblings and controls [30,40,48,50,57,58]. Longitudinal follow-up by Duytschaever et al. demonstrated a consistent lower temporal stability and reduced species abundance in CF patients compared to the control population [50]. The CF infants also had a reduced abundance of beneficial strains such as *Akkermansia, Eggerthella, Anostipes,* and *Clostridium IV* [37]. Further, there was a significant decrease in *Bacteroides*—a genus thought to be associated with immune modulation [55,59]. 

Early and prolonged exposure to antibiotics in patients with CF further exacerbates the alteration of the microbial composition [2,37]. Prominent butyrate producers such as *Anaerostipes, Butyricicoccus,* and *Ruminococcus* are reduced with antibiotic treatment in CF [37]. Also, the abundant Enterobacteriaceae species in CF exhibited resistance to amoxicillin compared to healthy subjects [50]. Antibiotic prescription practices were identified to contribute to this relative abundance of resistant strains among these patients [50]. Further investigations revealed that *E. coli* isolated from CF and controls revealed a differential gene expression. The CF isolates revealed a growth-promotional transcription profile, as opposed to a stress-associated, stationary-phase profile observed in control isolates [48].

### 3.3. Gut Dysbiosis with Altered Proteomics and Metabolomics in CF

Stool proteomic and metabolomic estimation techniques are helpful in further elucidating the functional aspects of dysbiosis [4]. Using 16S rRNA sequencing and metabolomics in stool samples from children with CF, Vernocchi and colleagues demonstrated abundant *Propionibacterium, Staphylococcus* and Clostridiaceae along with reduced abundance of *Eggerthella**, Eubacterium, Ruminococcus, Dorea, Faecalibacterium prausnitzii (F. prausnitzii)*, and Lachnospiraceae, and further noted that the resulting dysbiosis was associated with increased expression of metabolites such as gamma aminobutyric acid (GABA), choline, ethanol, propylbutyrate, and pyridine; and reduced levels of sarcosine, 4-methylphenol, uracil, glucose, acetate, phenol, benzaldehyde, and methylacetate [39] (Figure 1).

Utilizing liquid chromatography–mass spectrometry on stool protein extracts from CF patients and their healthy siblings as controls, Debyser et al. revealed the association of proteins intricately involved in inflammatory and mucus production pathways in CF patients [38]. The detailed microbial analysis revealed a decrease in butyrate reducing bacteria such as *F. prausnitzii* and an increase in Enterobacteriaceae, *Ruminococcus gnavus* and *Clostridia* species [38]. The stool metabolites between CF (in both pancreatic-sufficient and pancreatic-insufficient patients) and healthy controls differed significantly in abundance. A post hoc analysis revealed a decrease in butyric acid and pantetheine (which is needed for conversion of propanoate to propionyl coenzyme A) in CF [9,60]. Further, the investigators noted a high level of lipoyl-GMP, which could be utilized as a potential stool inflammatory marker [60]. The acidic intestinal milieu prevalent in CF may promote the growth of selective anaerobes such as *Clostridiales difficile* and *Propionibacterium,* which are known to produce metabolites such as alcohols, esters, and pyridine [39]. Wang et al. demonstrated that the commensal genus *Faecalibacterium* was strongly associated with SCFA production in healthy individuals [36]. This genus was substituted by *Clostridium sensu stricto* 1 in CF patients [36]. In a subset of CF patients, enterococcal overgrowth was noted, which was associated with increased lactate and reduced SCFA biosynthesis [36].

## 4. Impact of Gut Dysbiosis on Cystic Fibrosis Manifestations

### 4.1. Impact on the Gastrointestinal Tract

#### 4.1.1. Increase Intestinal Inflammation and Barrier Permeability

Alteration of the gut microbial milieu has been hypothesized as one of the contributors for intestinal inflammation and subsequent barrier impairment in CF [2,9,30,40,57,58,61,62,63,64] (Figure 1). Based on taxonomic and inferred functional dysbiosis prediction, children with CF had an increased abundance of pro-inflammatory pathogens such as *Staphylococcus**, Streptococcus, Escherichia**, Shigella, Enterobacter*, *Morganella,* and *Veillonella dispar,* along with decreased anti-inflammatory microbes like *Bacteroides*, *Bifidobacterium adolescentis*, and *F. prausnitzii* [9,30,33,37,40,48]. These findings are associated with ongoing intestinal inflammation as shown by the high level of fecal inflammatory markers [65,66,67,68,69]. Interestingly, the dysbiotic changes were similar to those reported in Crohn’s disease (CD), indicating a possibility of dysbiosis-mediated increased inflammation with intestinal permeability [59]. Additionally, in CF mice, decreased *Lachnoclostridium* and *Parabacteroides* were found to be associated with increased TH17 cells in the spleen, likely as a result of increased intestinal permeability [33]. The increased gut permeability in CF has been observed in both CF mouse models and human studies, and is attributed to many causes, including the CFTR dysfunction [63,70,71,72]. Essential fatty-acid deficiency, abnormalities in tight junctions, and decreased intestinal alkaline phosphatase are the other factors implied in increased intestinal permeability [63,70,71,72,73]. Intestinal alkaline phosphatase is an important mucosal defense factor necessary for gut homeostasis, and the low alkaline phosphatase activity in CF is secondary to increased release from the intestinal brush border, along with the faulty handling of this enzyme in post-Golgi compartments as a result of accumulation of the incorrectly glycosylated CFTR in these structures [63,70,74]. Thus the pattern of gut dysbiosis in CF may perpetuate intestinal inflammation and subsequent barrier impairment.

#### 4.1.2. Alteration of Fat Metabolism

Compared to healthy controls, patients with CF exhibited significantly decreased acetogens along with a reduction in butyrate producers such as *F. prausnitzii*, *Eubacterium rectale, Blautia spp*., *Ruminococcaceae,* and *Bacteroides,* resulting in substantial effects on host-cell metabolism [6,18,44,50,54,75]. Manor et al. demonstrated that children with CF had overall reduced potential in fatty-acid biosynthesis and an enhanced capacity for degrading butyrate and propionate compared to the healthy controls [30]. Similarly, Coffey et al., based on predicted functional analysis, reported an upregulation of the genes involved in the metabolism of propanoate and butyrate in CF patients [9,30]. Altered SCFA metabolism is noted in association with intestinal inflammation, and other metabolic effects such as poor bone growth [6,9,10,30]. Furthermore, functional dysbiosis is also associated with fat malabsorption, and as a result, the microbiome of CF patients are constantly exposed to high luminal fat content [30]. This may in turn contribute to the microbiome’s enhanced ability of fatty-acid metabolism [76]. The increased abundance of *E. coli* in CF patients was positively correlated with intestinal inflammation and altered lipid metabolism and absorption, thus potentially contributing to poor nutrition status [40]. (Figure 1)

#### 4.1.3. Gut Dysbiosis and Colon Cancer in CF

Colon cancer is about 5–10 times more prevalent in CF patients, and also occurs 2–3 decades earlier compared to the general population [77,78]. Gut inflammation is thought to play a key role in this predisposition to cancer in CF patients [41,78]. Along with intestinal inflammation, many other factors have been associated with this increased cancer predisposition, including the disruption of Wnt/β-catenin signal pathways, disturbances in gut stem-cell homeostasis, epithelial cell-junction disruptions, gut dysbiosis, and immune-cell infiltration [51,77] (Figure 1). *Fusobacterium*, widely associated with colorectal cancer, is observed to be also abundant in stool and colonic mucosal samples of patients with CF [9,51,79]. Dayama et al. demonstrated associations between gut dysbiosis, host–microbiome–gene interactions, and enrichment of oncogenic pathways [51]. Colonic mucosal RNA-seq was determined and 1543 expressed host genes were studied, along with an evaluation of mucosal dysbiosis using 16S rRNA sequencing [51]. Here, the authors reported the association of dysbiotic changes, like reduction in the butyrate producers *Ruminococcaceae* and *Butyricimonas*, and the increase in *Actinobacteria*, *Clostridium,* and the proinflammatory colorectal cancer-related microbiota *Veillonella* with the high expression of colorectal cancer-related genes such as lipocalin 2 (*LCN2*) and dual oxidase 2 (*DUOX2*) [51].

### 4.2. Extraintestinal Implications of Gut Dysbiosis in CF

#### 4.2.1. Gut Dysbiosis and Liver Involvement in CF

The role of gut dysbiosis in several liver disorders, such as nonalcoholic fatty liver disease, parenteral nutrition-associated liver disease, and primary sclerosing cholangitis, is increasingly recognized and referred to as a gut–liver axis [27,31,80]. Most common manifestations of liver involvement in CF include elevated liver enzymes and hepatic steatosis; the less common yet more severe presentation is termed as CF-related liver disease (CFLD) [80]. The exact pathophysiology of CFLD remains elusive. The conventional hypothesis includes CFTR dysfunction in the biliary epithelial cells, causing thick, tenacious bile leading to focal biliary cirrhosis, which may later progress to multifocal biliary cirrhosis [3,80]. Noncirrhotic portal hypertension with obliterative portal venopathy and nodular regenerative hyperplasia has been recently reported in children and young adults with CF, and their pathogenesis remains elusive [81].

Many investigators propose an alternate hypothesis for CFLD lying within the gut–liver axis model. Gut dysbiosis in CF may contribute to increased intestinal permeability, encouraging the entry of inflammatory mediators into the portal circulation and leading to activation of hepatic stellate cells [80]. Using video-capsule endoscopy, Flass and colleagues showed that the relative abundance of *Bacteroides* were reduced in CF patients with cirrhosis, which was associated with lower intestinal inflammation [82]. Also, they demonstrated that the *Clostridium* were more abundant in CF patients with cirrhosis, correlating with gut inflammation [82]. Additionally, CF patients with cirrhosis had slower transit, further underpinning the association with intestinal dysmotility, CFLD, and dysbiosis [82]. Debray et al. demonstrated CF-related biliary changes in *Cftr*−/− mice (both in congenic C57BL/6J and C57BL/6J;129/Ola haplotype backgrounds). These mice were randomly fed either with a medium-chain triglyceride (MCT)-based diet or chow with polyethylene glycol [3]. C57BL/6J *Cftr*−/− mice fed on an MCT-based diet developed biliary changes similar to CFLD, which correlated with dysbiosis, including *E. coli* preponderance, low-grade intestinal inflammation, enhanced gut permeability, and lack of secondary bile acids [3]. *Cftr*−/− mice fed with chow with polyethylene glycol did not develop cholangiopathy and also demonstrated a significant decrease in intestinal inflammation and concomitant decrease in *E. coli* [3]. The mixed C57BL/6J;129/Ola haplotype group did not develop cholangiopathy when fed with MCT; instead, they developed fatty-liver disease [3]. In CF mouse models, colitis was associated with biliary damage and portal inflammation, which was improved with antibiotics, further pointing to the possible role of gut-derived microbial products in the pathogenesis of CFLD [83] (Figure 1). Furthermore, when incubated with microbiota-derived metabolites like lipopolysaccharides, biliary epithelial cells secreted significantly more cytokines, which were mediated mainly by pro-inflammatory pathways like toll-like receptor-4 (TLR4) and NF-κB [83]. In mice models, the hepatic transcriptome profile in C57BL/6J *Cftr*−/− also showed increased expression of genes related to inflammation and reduced expression of genes related to immune tolerance [3]. Utilizing proteomics methodology, serum TIMP-4 and endoglin were noted to be increased in patients with CFLD, demonstrating a potential for a noninvasive biomarker for liver fibrosis [84]. However, this study did not evaluate the gut dysbiosis.

#### 4.2.2. Effect of Gut Dysbiosis on Growth Failure and Glucose Metabolism in CF Patients

Loman et al. reported that the relative abundance of *Staphylococcus* and *Faecalibacterium* species was negatively correlated with weight-for-length in young children with CF, while there was no significant association noted between alpha diversity with any anthropometric measurements, including body mass index Z scores [19]. Conversely, the beta diversity (between-sample diversity) varied significantly between normally growing and stunted CF populations [15,55]. Additionally, decreased length has been attributed to fecal microbial dysbiosis in infants with CF [26] (Figure 1). They also had decreased abundance of Bacteroidetes and a higher relative abundance of Proteobacteria compared to normal controls [26]. Further, 12/13 SCFA-producing species were significantly less pronounced in CF infants [26]. Dysbiosis has been extensively studied in diabetes, and similarly, CF-related diabetes also could primarily influence the gut dysbiosis, but needs further elucidation [13,85]. Investigators reported that the high relative abundance of *Alistipes* in CF patients was related to impaired glucose homeostasis, given its prominent role in succinate metabolism [9].

#### 4.2.3. Gut Dysbiosis and Respiratory Microbiome Interactions 

Patients with CF often have progressive obstruction of the airway with tenacious purulent secretions, which alter the respiratory microbiome. Similar to the intestinal microbiome, the microbial composition of the airway is also influenced by factors such as antibiotics and the severity of underlying disease process, which affects the mucosal rheology and leads to differential microbial colonization, and eventually dysbiosis [11]. In healthy cohorts, there is a continuum of the microbial flora from the upper to the lower respiratory tract [86]. However, this continuum is disrupted in CF patients, and a profound difference was noted in the diversity of flora between upper and lower airways, while demonstrating a significant correlation with the disease severity [87,88]. Proteobacteria and Actinobacteria are the dominant phyla in the CF lung microbiome [11]. Similar to gut dysbiosis, respiratory dysbiosis also impacts malnutrition, growth failure, and long-term respiratory and other CF comorbidities [89]. In CF, both lung and gut dysbiosis starts very early in life, and both could mutually impact each other [10,11,53,82]. Studies have shown a significant crossover between the respiratory microbiome and gut microbiome in CF patients with comparable changes in trend over time, highlighting a complex interplay between these two microbial populations [6,20,90]. The gut microbiome has been postulated to influence the airway microbiome via many mechanisms, such as the gut microbial gene products, including proteins and metabolites [10,11,90]. Similarly, microbial lung changes could impact the gut microbiome. For example, increased airway exacerbations in pediatric patients were positively correlated with gut dysbiosis [55]. Also, CF patients with severe lung dysfunction had significantly reduced gut microbiota diversity compared to patients with mild or moderate dysfunction [45]. This complex bidirectional influence is referred to as the gut–lung axis.

In a recent study, Antosca et al. demonstrated that intestinal dysbiosis is seen in children as early as six weeks of life, and this is associated with systemic inflammation and immune dysregulation leading to pulmonary exacerbations [55]. Additionally, GI dysbiosis induced by administration of nonabsorbable streptomycin treatment was associated with alterations in the pulmonary inflammatory cell profile, as well as airway hyperresponsiveness [91]. Burke et al. demonstrated a positive correlation between the diversity of the intestinal microbiome and the predicted forced expiratory volume in one second (FEV1) value [45]. On the contrary, in a recent study, no significant association was noted between the FEV1 or recent pulmonary exacerbation with the fecal microbiome species [19].

Studies on the lung microbiome have always dominated the landscape, but only few GI microbiome studies have shown association with clinical disease phenotypes and airway microbiome [92]. Hoen et al. and Madan et al. both documented an association between gut and respiratory microbiota in children with CF [20,53]. They found that a gut dysbiotic pattern characterized by pro-inflammatory microbiota like *Escherichia, Enterococcus, Parabacteroides* and *Blautia* over immunomodulatory genera like *Bacteroides* and *Bifidobacterium* [20,53]. These changes also preceded the pulmonary *Pseudomonas aeruginosa* colonization in this cohort [20,53]. Serial analyses of the gut and respiratory microbial samples in CF infants revealed an overlap between the two microbiota with a core microbiota dominated by *Veillonella* and *Streptococcus* [20]. The relative abundance of airway *Veillonella* is positively correlated with airway inflammation in CF patients [9,20,53]. Furthermore, the increased fecal *Streptococcus* abundance in children with intestinal inflammation appears to be specific to CF and convincingly argues for the hypothesis of the gut–lung axis [59]. The gut inflammation further perpetuates the intestinal-barrier impairment, resulting in the leakage of the pathogenic gut microbiome and their biologically active lipopolysaccharides to the circulation, which may explain the significant overlap with the respiratory microbiome and impact on the airways [2,64,93].

## 5. Methods to Modulate the Dysbiosis in Cystic Fibrosis

Various methods aimed at modulating the gut dysbiosis in CF include specific antibiotic therapy, dietary interventions, and FMT [9,89]. FMT is recommended for recurrent severe *Clostridioides difficile* infections, yet remains experimental for other indications. Potential nutritional strategies for modulating gut dysbiosis include modifications in macronutrient composition; supplementation of micronutrients; administration of probiotics, prebiotics (indigestible carbohydrates), or both (referred as synbiotics); and certain flavonoids [12,49,94]. (Figure 3 and Table 5).

### 5.1. Probiotics in Modulation of Gut Microbiome in CF

In their randomized control trial, Di Nardo et al. showed that supplementation with *Lactobacillus reuteri* ATCC55730 was associated with significantly decreased pulmonary exacerbations among CF patients with mild to moderate pulmonary involvement with the number needed to treat a value of 3 [95]. In addition to reducing pulmonary exacerbations, other studies have reported a significant decline in hospitalizations and improved quality of life [96,97]. The mechanisms of probiotic-mediated reduction of pulmonary exacerbations may include decreasing prevalence of potentially pathogenic microbiota such as Proteobacteria (Enterobacteriaceae), and *Fusobacterium* species and promotion of beneficial strains, such as *Lactobacillus* GG and *Bifidobacterium* species. *Lactobacillus* GG administration can reduce intestinal inflammation, as proven by decreases in calprotectin levels and increased digestive function after probiotic administration [56]. Similarly, *Lactobacillus*
*reuteri* supplementation was found to be associated with decreased fecal calprotectin and rectal nitric oxide levels, along with a trend toward restoring eubiosis among CF patients [56,61,98]. Additionally, probiotic intake was also reported to improve intestinal-barrier function [64]. Despite their rare and minimal adverse effects, at present, the routine prescription of probiotics in CF management is not recommended based on limited evidence [43,99]. Probiotic intervention studies in patients with CF are summarized in Table 6. 

### 5.2. Prebiotics in Modulation of Gut Microbiome in CF

Among available prebiotics, the oligosaccharides present in breast milk are relatively well studied. In healthy infants, breast milk oligosaccharides are associated with increased abundance of *Bifidobacterium* in an infant’s gut, playing a vital role in production of acetate and prevention of pathogenic infections such as *E. coli* [100]. Madan et al. noted that breast-milk feeding also was associated with increased diversity of gut microbiota and less respiratory tract colonization in CF infants [20]. Using an in vitro model, Wang et al. demonstrated the interplay between dysbiosis and prebiotics [36]. Here, levels of individual and total SCFA before and after high amylose maize starch fermentation by fecal slurries from CF and healthy participants were estimated [36]. Acetate and total SCFA levels were significantly reduced in CF patients compared to slurries from healthy control participants, while butyrate and propionate levels did not differ [36]. Taxa that showed positive correlations with the production of SCFAs in controls included *Dorea, Anaerostipes, Faecalibacterium,* and *Bacteroides,* and were significantly lower in postfermentation CF samples [36]. Acetate concentrations did not have any correlations among CF patient samples. Controls showed butyrate levels positively correlating with a relative abundance of *Faecalibacterium,* and that propionate was significantly associated with *Bacteroides* [36]. When CF fecal slurries were used, butyrate levels were strongly correlated with the *Clostridium sensu stricto* cluster 1 abundance, and propionate concentrations were significantly related to *Veillonella* fecal levels [36].

### 5.3. Effect of Vitamins and Dietary Nutrients on the Modulation of Gut Microbiome in CF 

The effect of vitamins on the modulation of gut microbiota has also been studied, though on smaller scale. Investigators have noted negative and positive associations between intake of antioxidant vitamins with *Bacteroides* and Firmicutes, respectively [101]. In a randomized study utilizing high-resolution metabolomics, high-dose vitamin D administration demonstrated an anticatabolic effect in adults with CF pulmonary exacerbation [102]. The simultaneous impact in gut microbiota was not evaluated. Long-term enrichment of polyunsaturated fatty acids has been shown to reduce pulmonary involvement and growth reduction in mice studies [103]. Further, mice supplemented with genistein, a soy isoflavone and polyphenol, were associated with reduced microbial diversity compared to a regular diet [104]. The dosage, duration, and long-term impact of administration of these nutrients or nutrient supplements to attain a healthy microbiota remains unknown [49].

### 5.4. Effect of Targeted Molecular Therapies on the Modulation of Gut Microbiome in CF

With the advent of highly effective targeted therapies in CF, evidence of their impact on restoring dysbiosis is emerging. In a small prospective observational study, a six-month course of ivacaftor, a CFTR potentiator, was associated with an increase in *Akkermansia* (bacterium involved in mucosal protection). Interestingly, the increased abundance of this genus negatively correlated with fecal inflammatory markers [105]. Kopp et al. reported favorable changes in serum metabolites involved in fat and amino-acid pathways after initiating lumacaftor/ivacaftor; however, correlations with the gut microbiome were not studied [106]. The effects of newer modulators such as lumacaftor/ivacaftor and lumacaftor/tezacaftor/ivacaftor on the gut microbiome are still uncertain, but may yield helpful insights into interactions between CF pathology and the microbial milieu.

## 6. Conclusions and Future Directions

With the advancements in science and technology, affordable and significant investigations have contributed substantial progress in understanding the role of the human microbiome in CF. As researchers explore the complex interactions between lung and gut microbial communities in the CF disease process, the focus will need to be directed toward potentially actionable interventions in an effort to manipulate the microbiome in beneficial ways starting early in life. Microbial richness, diversity, and dominant microbiota have the potential to be validated as biomarkers for the disease phenotype, and also can be helpful as prognostic factors [107]. More research is required to understand the dynamic interactions between microbiota and the variety of their subsequent metabolites to possibly identify therapeutic targets for metabolite-based therapy (postbiotics) toward personalized precision medicine [12]. Despite the increasing availability, multiomic technologies are still mostly available in research labs and are not utilized in clinical settings. There are numerous limitations of multiomics that need to be considered before routine clinical applications, including the high cost, lack of unique protocols, and challenges in choosing the right sample (e.g., stool microbiota may not be representative of all gut microbes, especially the mucosa-adherent microbiota) [17,108]. A large sample size is needed for validation cohorts to overcome false-positive test results [17,109,110].

Metatranscriptomics and metaproteomics offer the advantage of evaluating functional gene expression, but their utilization can be challenging [17]. Stool metatranscriptomics, which reads the transcribed gut mRNAs, are stable only for few minutes and may not entirely represent the in vivo expression [15,17]. Also, the transcribed RNA and expressed proteins may lack correlation limiting the interpretation of these modalities [17]. Metaproteomics, which utilizes liquid chromatography or gas chromatography coupled with mass spectrometry (LC/GC-MS), has the drawback of differentiating the origin of a protein (i.e., whether the expressed protein is from the host or is microbial in origin [110].

Similarly, metabolomics (targeted and untargeted), which also utilize LC/GC-MS, has its own limitations. Targeted metabolomics that include a pre-established set of non-protein molecules is preferred to the untargeted method, but not all the microbial molecules are predefined [17]. Untargeted metabolomics is limited by the inability to annotate all the molecules modified by microbial or host metabolism, and also possess the challenge of distinguishing the origin of molecules (host vs. microbial origin) [15,17]. The massive amount of data generated by these sophisticated methods also need advanced analytical computing [17,109,110].

Apart from bacterial microbiota, investigations into other microbial communities such as archaea, viruses (bacteriophages), fungi, and their interactions with the resident bacteriome, as well as the host, is another exciting area in need of more attention [30,111]. This is especially important in the context of emerging data that support an altered intestinal virome characteristic in CF patients [112]. Nevertheless, an ever-growing scientific investment in multiomic technology will provide promising opportunities to apply precision medicine strategies by designing more comprehensive and effective CF therapies in the future.

## Figures and Tables

**Figure 1 metabolites-11-00123-f001:**
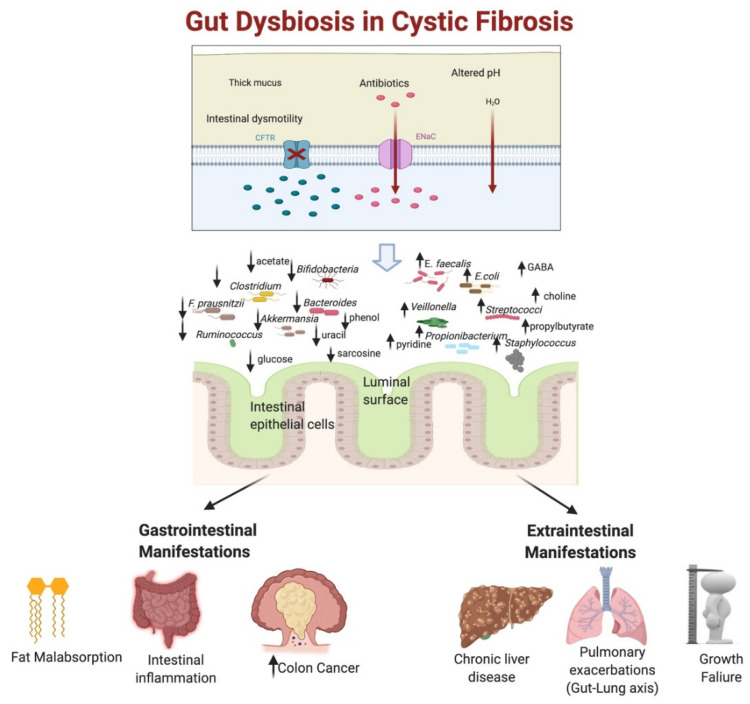
Patterns of gut dysbiosis in cystic fibrosis and their impact on the disease manifestations (created with BioRender.com). (↓—decrease; ↑—increase; *F. prausnitzii = Faecalibacterium prausnitzii; *E. faecalis** = *Enterococcus faecalis; E. coli* = *Escherichia coli*; GABA = gamma aminobutyric acid)

**Figure 2 metabolites-11-00123-f002:**
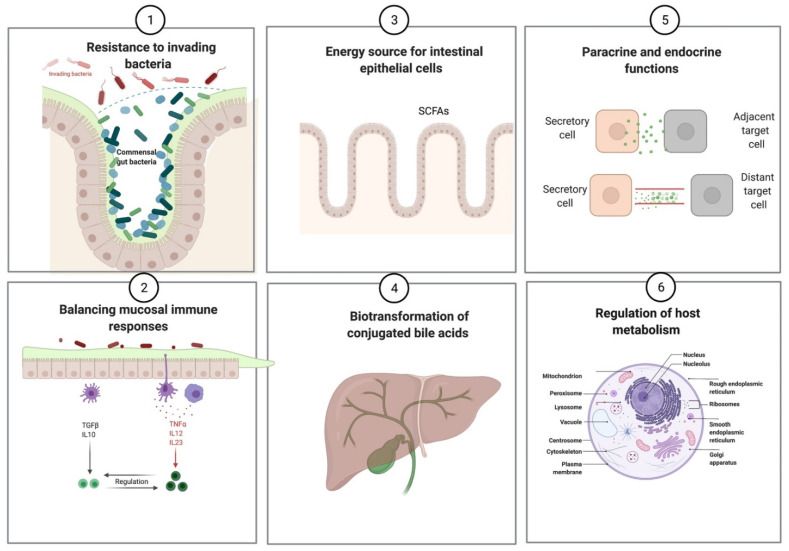
Functions of gut microbiota (Created with https://biorender.com/ (accessed on 18 February 2021)). (SCFAs = short-chain fatty acids).

**Figure 3 metabolites-11-00123-f003:**
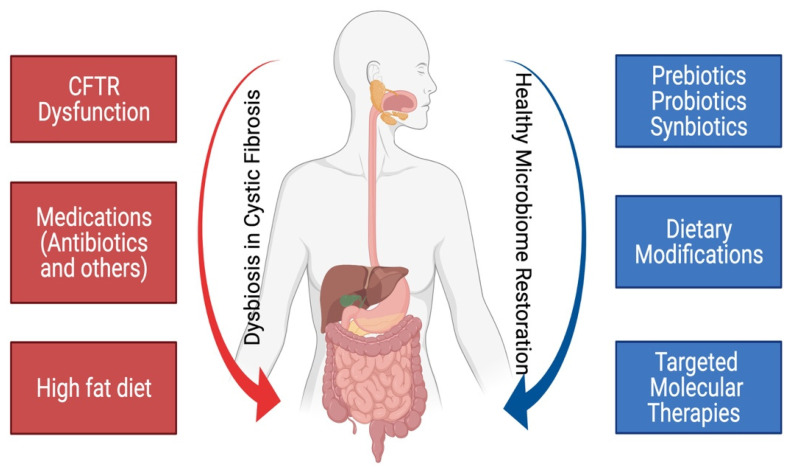
Gut dysbiosis in cystic fibrosis and therapies that aim to modulate the dysbiosis.

**Table 1 metabolites-11-00123-t001:** Functions of gut microbiota.

Prevention of pathogenic infections
Synthesis of vitamins (vitamin K and vitamin B complex)
Production of short-chain fatty acids Energy source for colonic enterocytes Maintenance of epithelial integrity and maturation Regulation of intestinal motility Amelioration of intestinal inflammation and immune signaling
Regulation of mucosal immune responses
Influence host metabolism and behavior via various endocrine and paracrine functions
Biotransformation of conjugated bile acids
Xenobiotic metabolism

**Table 2 metabolites-11-00123-t002:** Mechanisms of gut dysbiosis in cystic fibrosis.

**CFTR-related mechanisms**
Thick and inspissated mucus due to chloride channel dysfunction Defective bicarbonate secretion altering the intestinal milieu pH Malabsorption due to pancreatic insufficiency Intestinal dysmotility with prolonged intestinal transit Altered mucosal immune mechanisms Enhanced intestinal inflammation Epithelial barrier disruption
**Acquired factors**
Frequent use of antibiotics for recurrent pulmonary infections High-fat, high-calorie diet Other medications used in cystic fibrosis patients such as acid-suppressive medications, opioids, anticholinergic agents, and immunosuppressive mediations, etc.

**Table 3 metabolites-11-00123-t003:** Gut microbiome studies in cystic fibrosis.

**Study Population**	**Key Findings in CF**	**Conclusions**	**References**
7 infants with CF enrolled at birth	High degree of concordance between gut and respiratory microbial samples. For seven genera, gut colonization predicted their appearance in the lungs	Nutritional factors and gut colonization patterns could determine respiratory microbiome in CF	Madan et al., 2012 [20]
21 family units (one patient with CF and one to two healthy siblings)	↓ Abundance and temporal stability of *Bifidobacteria* and *Clostridium cluster* XIVa	Dysbiosis in CF could be due to disease-related impairment of essential gastrointestinal tract functions or a side effect of antibiotic usage	Duytschaever et al., 2013 [50]
12 patients with CF and 12 HC aged several weeks to 5 years	↑ *E. coli* correlated with nutrient malabsorption and intestinal inflammation	*E. coli* contributed to CF-related gastrointestinal dysfunction	Hoffman et al., 2014 [40]
13 patients with CF aged 0–34 months	Specific clustering of bacteria in fecal samples, but not respiratory samples, were associated with pulmonary exacerbations	Specific bacterial communities colonized the gut before the lungs in CF patients	Hoen et al., 2015 [53]
14 patients with CF and 12 HC aged < 3 years	↑ *E. coli, E. faecalis*, *Veillonella*, *C. difficile* ↓ *Beneficial Clostridiales* Dysbiosis significantly altered lipid metabolism (↓ FFA biosynthesis and ↑ anti-inflammatory SCFAs degradation)	Taxonomic and functional microbial shifts in young children with CF decreased with age Gut dysbiosis in CF correlated with fat malabsorption & inflammation	Manor et al., 2016 [30]
23 HC and 35 patients with CF (age range 0–18 years)	Progressive ↓ and alteration in richness and diversity of gut bacteria that was associated with CF from early childhood until late adolescence independent of pancreatic function	↑ Deviation in the number and diversity of intestinal microbiome with age in CF Efforts to rectify loss of bacterial diversity should be conducted no later than early childhood	Nielsen et al., 2016 [18]
43 patients with CF aged 21–38 years and 69 HC aged 24–40 years	↓ Microbial diversity ↑ Firmicutes ↓ Bacteroidetes	Gut dysbiosis in CF positively correlated with lung dysfunction and intravenous antibiotic use	Burke et al., 2017 [45]
30 patients with CF (14 were homozygous for delF508 and 14 were heterozygous, and 2 had mild genotype) age range 10–22 years and 8 HC (mean age 14.3 years)	↓ *Clostridium coccoides* ↓ *Bacteroides-Proveotella* ↓ *Bifidobacterium genera* ↓ Key butyrate producers	Low frequency of sulfate reducing bacteria in CF Significant reduction in hydrogen-consuming microbes in CF	Miragoli et al., 2017 [54]
31 patients with CF between 1–6 years and age-matched 1:1 HC	↑ *Propionibacterium, Staphylococcus, C. difficile* ↓ *Eggerthella, Eubacterium, Ruminococcus, F. prausnitzii, Lachnospiraceae* ↑ GABA, choline, propylbutyrate, and pyridine↓ Sarcosine, methylphenol, uracil, glucose, acetate, phenol, and benzaldehyde	CF gut microbiota revealed an enterophenotype that was correlated with disease status regardless of age and pancreatic status. This distinct dysbiosis was partially related to pulmonary infections and oral antibiotic use	Vernocchi et al., 2018 [39]
27 patients with CF and age/gender matched HC (age range 0.8–18 years)	Prominent taxonomic and functional dysbiosis in CF compared to HC ↓ richness and diversity of gut microbiota in CF	Enrichment of genes involved in SCFAs, antioxidant and nutrient metabolisms in CF	Coffey et al., 2019 [9]
21 patients with CF and 409 healthy infant controls	Unlike the healthy infants, the alpha diversity did not increase in CF infants ↓ *Bacteroides* ↓ *Roseburia* ↑ *Veillonella*	The distinct CF gut microbiota in infants was associated with pulmonary exacerbations. In vitro models suggested the role of *Bacteroides* in reduction of IL-8 linking the gut dysbiosis in CF-related inflammation	Antosca et al., 2019 [55]
20 patients with CF and 45 HC, fecal samples collected over the first 18 months of life	↓ *Akkermansia, Bifidobacterium, Bacteroides* and *Anaerostipes* ↑ *Streptococci, Enterococcus* and *E. coli* ↓ Alpha diversity	Antibiotic use in infants with CF was associated with a lower alpha diversity and altered microbial composition	Kristensen et al., 2020 [37]
207 infants with CF and 25 HC	↓ Bacteroidetes ↑ Proteobacteria	CF infants with low length had pronounced dysbiosis than HC and CF infants with normal length	Hayden et al, 2020 [26]

(CF = cystic fibrosis; ↓—decrease; ↑—increase; HC = healthy controls; *E. coli* = *Escherichia coli; E. faecalis* = *Enterococcus*
*faecalis*; *C. difficile* = *Clostridiales difficile*; FFA = free fatty acids; SCFAs = short-chain fatty acids; *F. prausnitzii = Faecalibacterium prausnitzii*; GABA = gamma aminobutyric acid; IL-8 = interleukin-8).

**Table 4 metabolites-11-00123-t004:** Features of gut dysbiosis in cystic fibrosis.

	Increase	Decrease
**Phylum level**	Firmicutes/Bacteroidetes ratio γ-Proteobacteria	Bacteroidetes Firmicutes Actinobacteria
**Genus/Species level**	Pro-inflammatory microbiota *Enterobacteriaceae* *Streptococcus* *Veillonella* *Staphylococcus* *Propionibacterium* (promoted by low pH and anaerobic milieu) Colorectal cancer-related microbiota *Fusobacterium* *Veillonella* *Escherichia coli* (growth–promotional transcription profile) Antibiotics-resistant species *Enterococcus faecalis* *Enterococcus faecium* Pro-inflammatory species *Veillonella dispar* *Clostridiales difficile*	Beneficial microbiota *Bifidobacterium* *Clostridium* *Akkermansia* *Eggerthella* Immune modulatory microbiota *Bacteroides species* Butyrate producers *Anaerostipes* *Butyricicoccus* *Ruminococcus* *Faecalibacterium prausnitzii* *Eubacterium rectale* *Blautia species* Anti-inflammatory species *Bifidobacterium adolescentis*
**Metabolite level**	Gamma aminobutyric acid Choline Ethanol Propylbutyrate Pyridine	Butyric acid Pantetheine Sarcosine Methylphenol Uracil Glucose Acetate Phenol Benzaldehyde Methylacetate

**Table 5 metabolites-11-00123-t005:** Therapies that aim to modulate the effect of gut dysbiosis in cystic fibrosis.

Specific antibiotic therapy
Dietary interventions (modification of macronutrient composition)
Probiotics
Prebiotics (indigestible carbohydrates)
Probiotics and prebiotics (synbiotics)
Vitamins and supplements
Fecal microbiome transplantation
Targeted molecular therapies (e.g., ivacaftor)
Metabolite-based therapies (postbiotics)

**Table 6 metabolites-11-00123-t006:** Probiotic intervention studies in cystic fibrosis.

Study Population	Probiotic Strains	Clinical Responses	Proposed Mechanisms	References
19 patients	*LGG**	↓ Pulmonary exacerbations and hospital admissions	↓ DCs* maturation resulting in induction of Treg*-cells	Bruzzese et al., 2007 [96]
37 patients (20 received probiotics and 17 took placebo capsules)	*L. casei**, *L. rhamnosus**, *S. thermophiles**, *B. breve**, *L. acidophilus**, *B. infantis**, *L. bulgaricus**	↓ Pulmonary exacerbations and improving quality of life	Preventing deleterious effects of inflammatory cytokines (TNF-α * IFN-γ*) on epithelial function leading to a less-disrupted intestinal barrier	Jafari et al., 2013 [97]
61 Patients with mild to moderate pulmonary disease	*L. reuteri* * ATCC55730	↓ Pulmonary exacerbations and URTI*	Improvement of intestinal barrier function and modulation of immune response	Di Nardo et al., 2014 [95]
22 patients aged 2–9 years	*LGG**	↓ Fecal calprotectin (↓ intestinal inflammation)	Partial restoration of healthy intestinal microbiota that limit intestinal inflammation	Bruzzese et al., 2014 [61]
30 patients in two groups (probiotic and placebo group)	*L. reuteri**	↓ Fecal calprotectin (↓ intestinal inflammation) and ↑ digestive comfort	↑ Microbial diversity with ↑ representation of Firmicutes. ↓ γ-Proteobacteria genera Enterobacteriaceae	Del Campo et al., 2014 [56]
25 patients aged 7–12 years (crossover study)	*L. rhamnosus** SP1 (DSM 21690) & *B. animalis* spp.BLC1 (LGM23512)	Normalization of gut permeability in 13% of patients. No change in BMI*, FEV1*%, abdominal pain, and pulmonary exacerbations	Probiotic supplementation did not change the microbiota (both at phylum or phylogenetic levels)	Van Biervliet, 2018 [64]

(**LGG* = *Lactobacillus GG*; DCs = dendritic cells; Treg = regulatory T-cells; *L. casei = Lactobacillus casei*; *L. rhamnosus = Lactobacillus rhamnosus*; *S. thermophilus* = *Streptococcus thermophiles;*
*B. breve =*
*Bifidobacterium breve*; *L. acidophilus =*
*Lactobacillus acidophilus*; *B. infantis = Bifidobacterium infantis*; *L. bulgaricus = Lactobacillus bulgaricus*; TNF-α *=* tumor necrosis factor alpha; IFN-γ = interferon gamma; *L. reuteri = Lactobacillus reuteri*; URTI = upper respiratory tract infections; *B. animalis = Bifidobacterium animalis*; BMI = body mass index; FEV1 = forced expiratory volume 1; ↓—decrease; ↑—increase).

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
