# Peer review of "Impact of Altered Gut Microbiota and Its Metabolites in Cystic Fibrosis"

_metabolites, 2021, doi:10.3390/metabo11020123_

Round 1

Reviewer 1 Report

When we talk about Cystic Fibrosis, we always talk about either CFTR or colonization of the airways by Pseudomonas aeruginosa, without giving worthy space to the metabolic disorders that the CF cell faces, nor to the repercussions of the disease on other organs not directly affected. Therefore, I believe that the argument itself gives credit to the article for scientific soundness and novelty in the field of CF research.
That Microbial diversity is strongly correlated with health is certainly a cutting-edge topic on the scientific research scenario, not only CF.
Therefore the subject in question is really very interesting, even more so if, as the authors say, gastrointestinal manifestations in CF may begin in utero and continue throughout the life resulting in a chronic state of an altered intestinal milieu.
Article is well-written and well-organized, fluid and interesting to read, the different arguments balanced and explored in the specific Sections and a good job of synthesizing the literature, including very recent discoveries, has been made.
I am of the opinion that the author answer the questions he / she sets out to answer in a comprehensive and satisfactory way.
Only one observation (Minor point) is suggested to improve the paper and make it 'accessible' to readers outside the area of interest of the topic: Figure 1 should be more highlighted in the text and surely more figures would help summarizing / visualizing all the data discussed in the text.

Author Response

Reviewer 1

Reviewer comments:

When we talk about Cystic Fibrosis, we always talk about either CFTR or colonization of the airways by Pseudomonas aeruginosa, without giving worthy space to the metabolic disorders that the CF cell faces, nor to the repercussions of the disease on other organs not directly affected. Therefore, I believe that the argument itself gives credit to the article for scientific soundness and novelty in the field of CF research. That Microbial diversity is strongly correlated with health is certainly a cutting-edge topic on the scientific research scenario, not only CF. Therefore the subject in question is really very interesting, even more so if, as the authors say, gastrointestinal manifestations in CF may begin in utero and continue throughout the life resulting in a chronic state of an altered intestinal milieu. Article is well-written and well-organized, fluid and interesting to read, the different arguments balanced and explored in the specific Sections and a good job of synthesizing the literature, including very recent discoveries, has been made. I am of the opinion that the author answer the questions he / she sets out to answer in a comprehensive and satisfactory way.

Author’s response:

We sincerely thank you for your time and appreciate your comments.

Reviewer comments:

Only one observation (Minor point) is suggested to improve the paper and make it 'accessible' to readers outside the area of interest of the topic: Figure 1 should be more highlighted in the text and surely more figures would help summarizing/visualizing all the data discussed in the text.

Author’s response:

Thank you again for this suggestion. We do agree and have now incorporated Figure 1 (red font) at multiple places in the text. Also, we added two more figures (Figures 2 and 3) as recommended to help the readers summarize the text data.

Reviewer 2 Report

Thavamani et al. conducted a comprehensive review about the interactions between Gut Microbiota, Nutrition, and Metabolism in Cystic Fibrosis in this study, which covered materials in many studies and wide topics in CF. However, this study has an obvious limitation: there are little discussion of the mechanisms in CF on their disfunction or health associated with Gut Microbiota, Nutrition, and Metabolism.

Additionally, the  conclusion is simple and there is no discussion of  limitations of current methods, including technique and methodology.

Author Response

 Reviewer 2

Thavamani et al. conducted a comprehensive review about the interactions between Gut Microbiota, Nutrition, and Metabolism in Cystic Fibrosis in this study, which covered materials in many studies and wide topics in CF. However, this study has an obvious limitation: there are little discussion of the mechanisms in CF on their dysfunction or health associated with Gut Microbiota, Nutrition, and Metabolism.

Author’s response:

We sincerely thank you for your time and appreciate your comments. In the revised version, the mechanisms in CF resulting in dysbiosis are discussed in sections 3 and 4 and also highlighted in Table 2. If this reviewer considers that the title is not in align in with the review, we would like to change the title to “Impact of Altered Gut Microbiota and its Metabolites in Cystic Fibrosis”

Reviewer comments:

Additionally, the conclusion is simple and there is no discussion of limitations of current methods, including technique and methodology.

Author’s response:

In the conclusion section, we have added a section discussing the limitations of current methods focusing on methodology and technique as suggested and also added more appropriate references. Thank you again.

Round 2

Reviewer 2 Report

The authors have added discussion of the mechanisms in CF and discussed the limitations of current approaches.